# Decoupled maternal and zygotic genetic effects shape the evolution of development

Christina Zakas*, Jennifer M Deutscher, Alex D Kay, Matthew V Rockman*

Center for Genomics & Systems Biology, Department of Biology, New York University, New York, United States

**Abstract** Evolutionary transitions from indirect to direct development involve changes in both maternal and zygotic genetic factors, with distinctive population-genetic implications, but empirical data on the genetics of such transitions are lacking. The polychaete *Streblospio benedicti* provides an opportunity to dissect a major transition in developmental mode using forward genetics. Females in this species produce either small eggs that develop into planktonic larvae or large eggs that develop into benthic juveniles. We identify large-effect loci that act maternally to influence larval size and independent, unlinked large-effect loci that act zygotically to affect discrete aspects of larval morphology. The likely fitness of zygotic alleles depends on their maternal background, creating a positive frequency-dependence that may homogenize local populations. Developmental and population genetics interact to shape larval evolution.

DOI: https://doi.org/10.7554/eLife.37143.001

*For correspondence:
cz12@nyu.edu (CZ);
mrockman@nyu.edu (MVR)

**Competing interests:** The authors declare that no competing interests exist.

## Introduction

Many animals develop indirectly via larval stages that are morphologically and ecologically distinct from their adult forms. Hundreds of lineages across animal phylogeny have secondarily lost larval forms, instead producing offspring that directly develop into adult form without a distinct larval ecological niche (*Raff, 1996*; *Smith et al., 2007*; *Strathmann, 1985*; *Thorson, 1950*). Indirect development in the sea is typically planktotrophic: females produce large numbers of small offspring that require exogenous planktonic food to develop before metamorphosing into benthic juveniles. Direct development is typically lecithotrophic: females produce a smaller number of larger eggs, each developing into a juvenile without the need for larval feeding, provisioned by yolk. Evolutionary theory suggests that these alternative developmental strategies represent stable alternative fitness peaks, while intermediate states are disfavored (*Christiansen and Fenchel, 1979*; *Havenhand, 1995*; *Strathmann, 1985*; *Vance, 1973*). Transitions from planktotrophy to lecithotrophy thus represent radical and coordinated transformations of life-history, fecundity, ecology, dispersal, and development (*Duda and Palumbi, 1999*; *Jeffery and Emlet, 2003*; *McEdward and Miner, 2001*; *Raff, 1996*; *Romiguier et al., 2014*; *Wray, 1995*). Here we dissect this transition in the polychaete annelid *Streblospio benedicti*, a genetically tractable species that harbors both states as heritable variation (*Levin, 1984*; *Levin et al., 1991*; *Zakas and Rockman, 2014*).

*Streblospio benedicti* is a common and widespread benthic polychaete found in estuaries throughout North America. Females brood fertilized embryos in dorsal brood pouches and release larvae into the water column. Larvae remain pelagic until they are competent to metamorphose and return to the benthos. Females fall into two classes exhibiting classic life-history trade-offs: planktotrophic mothers produce small (~100 um) eggs that develop into obligately feeding larvae that spend weeks in the plankton. These larvae grow larva-specific swimming chaetae that are thought to deter predation (*Blake, 1969*) (*Figure 1A*). Lecithotrophic mothers produce large (~200 um) eggs

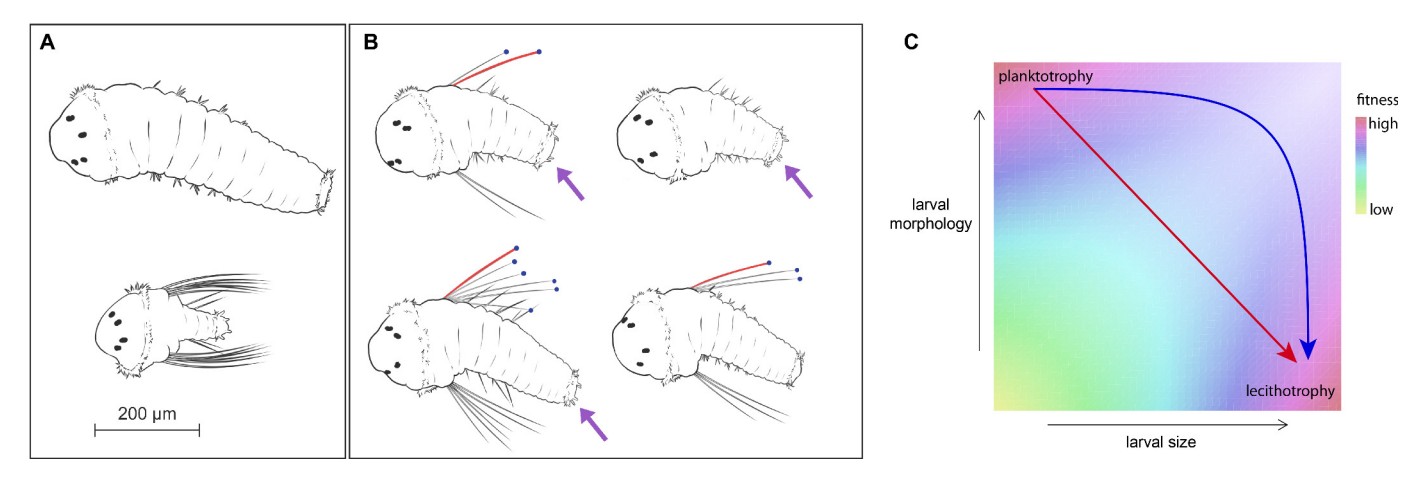

**Figure 1.** *S. benedicti* larval morphology. (**A**) Wild populations occupy two extremes of larval size and form: planktotrophic larvae (bottom) carry larva-specific chaetae and anal cirri, while lecithotrophic larvae (top) lack these traits and are capable of adopting a benthic habit without feeding. (**B**) $G_2$ animals (right panel) are intermediate in size and variable in their larval morphologies. We measured larval area, chaetae length (red), presence of anal cirri (purple arrows) and number of chaetae per side, when present (blue dots). (**C**) Evolutionary theory and the absence of intermediates in the wild suggest a hypothetical fitness landscape, in which planktotrophs and lecithotrophs occupy fitness peaks separated by a valley in size-shape-fitness space (*Christiansen and Fenchel, 1979*; *Havenhand, 1995*; *Strathmann, 1985*; *Vance, 1973*). Transitions from planktotrophy to lecithotrophy may involve single-step pleiotropic transformations (red arrow) or independent evolution of size and form (blue arrow).

DOI: https://doi.org/10.7554/eLife.37143.002

that develop into larvae that do not require exogenous food and quickly leave the water column as benthic juveniles. These larvae lack swimming chaetae and a second type of larva-specific morphological structure, anal cirri containing bacillary cells, which are distinctive cells that contain rhabdites of unknown function (*Gibson et al., 2010*) (*Figure 1A*). There are also differences during embryogenesis (*McCain, 2008*) and organogenesis (*Gibson et al., 2010*; *Pernet and McHugh, 2010*), where accelerated development of juvenile features and truncated development of larval features occurs in lecithotrophs. Despite these developmental and larval differences, adults are indistinguishable at the level of gross morphology. There are no known behavioral differences between the adults, though it is possible that preferences in mating or habitat choice occur. Unlike examples of polyphenism, environmental factors do not substantially alter egg size or subsequent offspring type (*Levin and Bridges, 1994*; *Levin and Creed, 1986*). Natural populations consist of only one of the two phenotypic extremes, often occurring in close geographic proximity but rarely directly overlapping (*Levin, 1984*; *Schulze et al., 2000*; *Zakas and Wares, 2012*), and larval mode in the population does not correlate with any known environmental gradient. Despite the persistence of these two dramatically different strategies in nature, they can be crossed in the lab producing an array of morphological and developmental intermediates (*Levin et al., 1991*; *Zakas and Rockman, 2014*; *Figure 1B*). Stable populations of different larval modes suggest that migration-selection balance on small ecological scales is acting to maintain both types in the larger population.

We genetically dissected developmental mode by crossing a lecithotrophic male from Long Beach, California, and a planktotrophic female from Bayonne, New Jersey. Studies of these populations have found them to be homogeneous in larval mode but little differentiated from one another across the genome (*Rockman, 2012*; *Zakas and Rockman, 2014*; *Zakas and Rockman, 2015*), consistent with ongoing gene flow between larval modes in their native East Coast range (*Zakas and Wares, 2012*). West Coast populations, including Long Beach, are exclusively lecithotrophic and result from a 20th century introduction from an East Coast source (*Zakas and Wares, 2012*).

Genetic models for evolutionary transitions from planktotrophy to lecithotrophy require traversal of intermediate states in multivariate phenotype space, which are predicted to be suboptimal with respect to fitness (*Vance, 1973*; *Whitlock et al., 1995*) (*Figure 1C*). The simplest scenario is that coadapted trait combinations are affected by shared alleles, either by pleiotropy (for example, ref [*Hall et al., 2006*]) or tight linkage (e.g., a supergene with suppressed recombination [*Jones et al.,*

2012; *Joron et al., 2011*; *Yeaman, 2013*]). The necessity to coordinate maternal traits (larval size) with zygotic traits (larval morphology) adds additional constraints (*Kirkpatrick and Lande, 1989*; *Wade, 1998*; *Wolf and Brodie, 1998*; *Wolf and Wade, 2016*). *Figure 1C* shows that alternative paths from planktotrophy to lecithotrophy make different predictions about the correlations among genetic effects. In particular, the presence of facultative planktotrophy (i.e., larvae can feed but do not require food) in lecithotrophic *S. benedicti* larvae (*Allen and Pernet, 2007*; *Pernet and McArthur, 2006*) provides one suggestion that evolution could traverse the larval-size axis, from small to large, prior to traversing the larval morphology axis, from 'complex' feeding larvae to 'simple' non-feeding larvae with accelerated development of juvenile features (*Smith et al., 2007*; *Wray, 1996*). Nevertheless, in natural populations we observe only the two extremes: large, 'simple' lecithotrophs, and small, 'complex' planktotrophs.

## Results

From the $F_1$ progeny of our cross between a lecithotroph and planktotroph, we generated a $G_2$ panel by crossing one $F_1$ male with four $F_1$ females. We measured larval features – size, number and length of swimming chaetae, and presence or absence of anal cirri – on each $G_2$ animal, then raised it to adulthood and crossed it to another $G_2$ animal. From the progeny of these crosses, we measured average size of each $G_2$ female's $G_3$ offspring. This extra generation is required because larval size is a maternal-effect trait (*Zakas and Rockman, 2014*). Indeed, we find that the larval size of a $G_2$ female is uncorrelated with the average size of her $G_3$ progeny (*Figure 2A*). The $G_2$ larvae all develop with equivalent maternal genetic effects, as their $F_1$ mothers are heterozygotes for planktotrophic and lecithotrophic alleles. By contrast, maternal-effect genes are segregating among $G_2$ females, and so $G_3$ larval sizes reflect inherited maternal-effect variation.

Next we constructed a genetic map, the first for any annelid (*Supplementary file 1*). We genotyped $F_1$ and $G_2$ animals at 702 markers, which fell into 10 autosomal and one sex-linked linkage group (LG) (*Broman et al., 2003*; *R Core Development Team, 2015*). In parallel, we defined the karyotype for *S. benedicti*, which includes 10 pairs of autosomes and one pair of sex chromosomes (*Figure 2B–D*, *Figure 2—figure supplement 1*). Males are heterogametic, diagnosed by a large Y chromosome. The karyotypes are grossly indistinguishable between Long Beach and Bayonne, and in each population we observe 11 bivalents at meiotic prophase in females and 12 bodies, presumably 10 autosomal bivalents and two sex-chromosome univalents, at diakinesis in males (*Figure 2C, D*; *Figure 2—figure supplement 1*).

$G_2$ larvae are intermediate in size and variable in morphology; 15% lack larval chaetae and 25% lack anal cirri. As expected for a maternal trait, $G_2$ larval size showed no linkage to $G_2$ genotype. Every other phenotype implicated a small number of large-effect loci (*Figure 2E*; *Table 1*). Moreover, the loci are almost entirely independent: the maternal-effect loci shaping $G_3$ larval size are unlinked to the zygotic-effect loci shaping $G_2$ larval morphology. The morphology loci are also largely independent, with a major-effect anal cirri locus on LG5 and loci uniquely affecting number and length of swimming chaetae on LG8 and LG9. The two chaetal traits also share a common locus on LG3. Each QTL has a large effect (*Figure 3A*) and explains a large proportion of trait variance (*Table 1*), and the effects are largely additive (*Table 2*). The two exceptions to additivity are in chaetae length at the LG3 locus (the chaetae-shortening allele is dominant) and larval size, where there is significant epistasis between the two maternal-effect loci on LG6 and LG7. We corroborated the single-trait mapping results by performing a fully multivariate mapping analysis for maternal-effect larval size, chaetae number, and chaetae length. This analysis identified the same loci as the univariate analyses, and showed that the maternal effects and zygotic effects are largely independent (i.e., most effects are nearly orthogonal in this three-dimensional space: *Figure 3B*).

Our results imply that matings between animals of alternative developmental modes will routinely generate mismatches between larval size and morphology. This genetic architecture creates a strong selective barrier to adaptation: zygotic-effect alleles that migrate into new populations will find themselves tested against the local maternal-effect background (*Kirkpatrick and Lande, 1989*; *Wolf and Brodie, 1998*; *Wolf, 2000*). This maternal-by-zygotic epistasis for fitness can inhibit adaptation to local environmental conditions by creating runaway coevolution (*Wade, 1998*): the locally common maternal background creates a selective regime favoring matching zygotic-effect alleles.

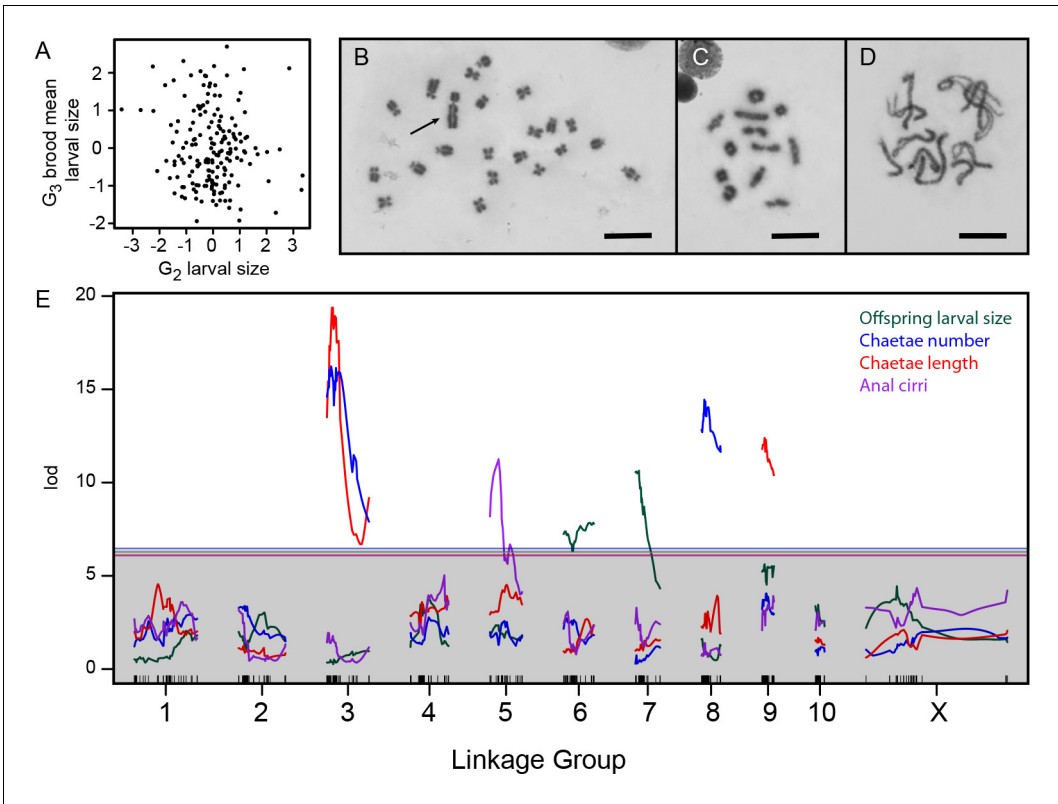

**Figure 2.** Genetics of developmental mode. (**A**) Larval size is uncorrelated between the $G_2$ and $G_3$ generations ($r$ = –0.1, p=0.16), consistent with maternal inheritance. (**B**) The 22 chromosomes of a male from the Bayonne population include a large Y (arrow). Scale bar is 10 μm in panels B-D. (**C**) Meiotic chromosomes at diakinesis in the same animal show 12 objects, putatively 10 autosomal bivalents and two sex-chromosome univalents. (**D**) Zygotene in a Bayonne female shows 11 bivalents. (**E**) Interval mapping of four traits in the $G_2$ population identifies seven linkages on six chromosomes. Thresholds for genome-wide significance at p=0.05 are indicated for each trait by horizontal lines, determined by 1000 structured permutations that account for $G_2$ family structure.

DOI: https://doi.org/10.7554/eLife.37143.003

The following figure supplement is available for figure 2:

**Figure supplement 1.** Karyotypes.

DOI: https://doi.org/10.7554/eLife.37143.004

**Table 1.** $G_2$ phenotypic variance explained by significant loci and interactions.

| Trait | Locus | Variance explained* |
|---|---|---|
| Number of chaetae | LG3 | 21.8% |
| | LG8 | 18.2% |
| Length of chaetae | LG3 | 25.8% |
| | LG9 | 13.5% |
| $G_3$ mean larval size | LG6 | 18.8% |
| | LG7 | 25.4% |
| | LG6 x LG7 | 5.2% |
| Presence of anal cirri | LG5 | 18.9% |

*Percent variance explained is estimated by dropping the specified locus or interaction from the best-fitting genetic model for the phenotype (File S4). In the case of anal cirri, the reported number is the percent deviance explained in a logistic regression.

DOI: https://doi.org/10.7554/eLife.37143.005

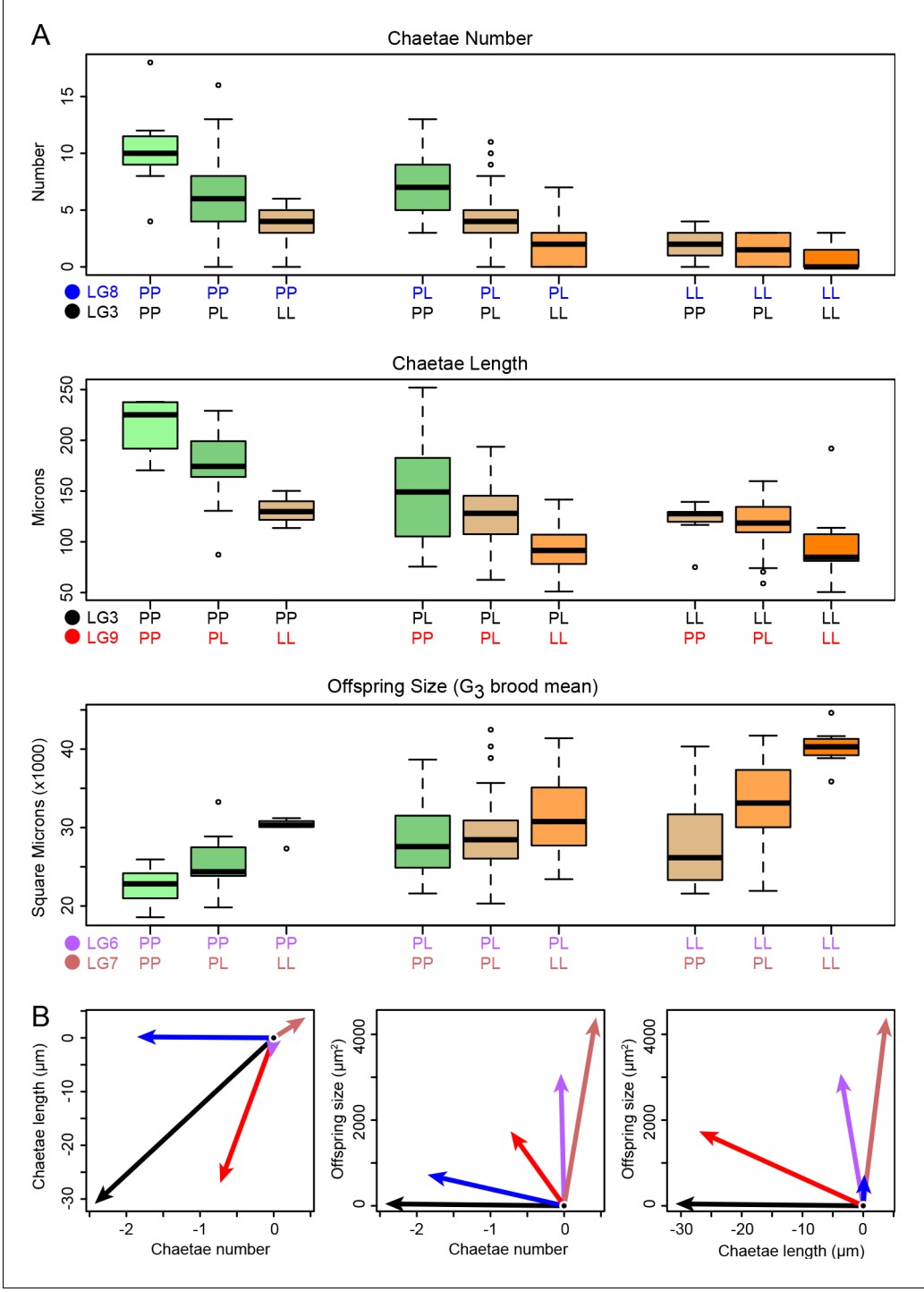

**Figure 3.** Phenotype distributions for each genotype. (**A**) Loci have substantial and largely additive effects. Each boxplot shows the phenotype distribution as a function of the specified $G_2$ two-locus genotype, and each box is colored according to the number of planktotroph (P) and lecithotroph (L) alleles it carries. The horizontal bar shows the median, the box spans the interquartile range, the whiskers encompass all data within 1.5 times the interquartile range, and points beyond this range are shown individually. (**B**) Multivariate analysis shows that the effects of the five loci in panel A are largely restricted to maternal or zygotic traits. The additive-effect vectors for lecithotrophic alleles in three-trait space are here projected onto two dimensions. Arrow color corresponds to the locus colors in panel A. The major maternal-effect loci are strikingly nearly orthogonal to the major zygotic effects (LG3 and LG8), while the locus on LG9 is mildly pleiotropic, with major effects on chaetae length and minor effects on number and offspring size (all in the aligned direction: shorter chaetae, fewer chaetae, larger offspring).

*Figure 3 continued on next page*

*Figure 3 continued*

DOI: https://doi.org/10.7554/eLife.37143.007

This frequency-dependent runaway model assumes that zygotic-effect alleles are penetrant across maternal-effect backgrounds. An alternative is that zygotic effects are masked in mismatched maternal backgrounds, allowing them to persist invisibly until their matching maternal background becomes available. To test this possibility, we backcrossed 30 $G_2$ males (which, like all of the $G_2$s, have an $F_1$ maternal background) to planktotrophic females, thereby shifting the maternal background to assess the penetrance of the chaetae loci. We found that the zygotic-effect loci that affect chaetae length and number in an $F_1$ maternal background also do so in the planktotroph maternal background (length: $p<10^{-4}$; number $p<10^{-6}$; *Figure 4*). We estimate a negligible effect of the locus on LG9, but our analysis of that locus is underpowered due to unbalanced genotype frequencies among the 30 $G_2$ males.

**Table 2.** Effect sizes for significant QTL for each trait.

| Locus | Effect | Effect size | SE |
|---|---|---|---|
| Chaetae Number | | number | |
| Intercept | | 4.29 | 0.18 |
| LG3 2.5 cM | Additive | −2.53 | 0.27 |
| LG3 2.5 cM | Dominance | −0.58 | 0.35 |
| LG8 1.7 cM | Additive | −2.30 | 0.27 |
| LG8 1.7 cM | Dominance | 0.31 | 0.35 |
| Chaetae Length* | | μm | |
| Intercept | | 140.02 | 5.02 |
| Family A | | 2.60 | 6.53 |
| Family C | | 0.01 | 6.09 |
| Family H | | −19.72 | 6.04 |
| LG3 3.6 cM | Additive | −29.59 | 3.17 |
| LG3 3.6 cM | Dominance | −20.26 | 4.06 |
| LG9 1.5 cM | Additive | −25.07 | 3.61 |
| LG9 1.5 cM | Dominance | 4.11 | 4.33 |
| $G_3$ Offspring Area | | μm$^2$ | |
| Intercept | | 29516.79 | 352.34 |
| LG6 17.9 cM | Additive | 3356.14 | 498.74 |
| LG6 17.9 cM | Dominance | −908.67 | 704.69 |
| LG7 2.1 cM | Additive | 4020.69 | 546.01 |
| LG7 2.1 cM | Dominance | −228.43 | 704.69 |
| LG6 x LG7 | AxA | 1254.06 | 821.20 |
| LG6 x LG7 | DxA | 232.10 | 1092.04 |
| LG6 x LG7 | AxD | −3297.03 | 997.48 |
| LG6 x LG7 | DxD | 62.26 | 1409.39 |
| Anal Cirri | | logistic | |
| Intercept | | 3.85 | 26.57 |
| LG5 5.1 cM | Additive | −6.68 | 53.14 |
| LG5 5.1 cM | Dominance | −6.07 | 53.14 |

*This model includes a family effect, coded with Family F as the reference family

DOI: https://doi.org/10.7554/eLife.37143.006

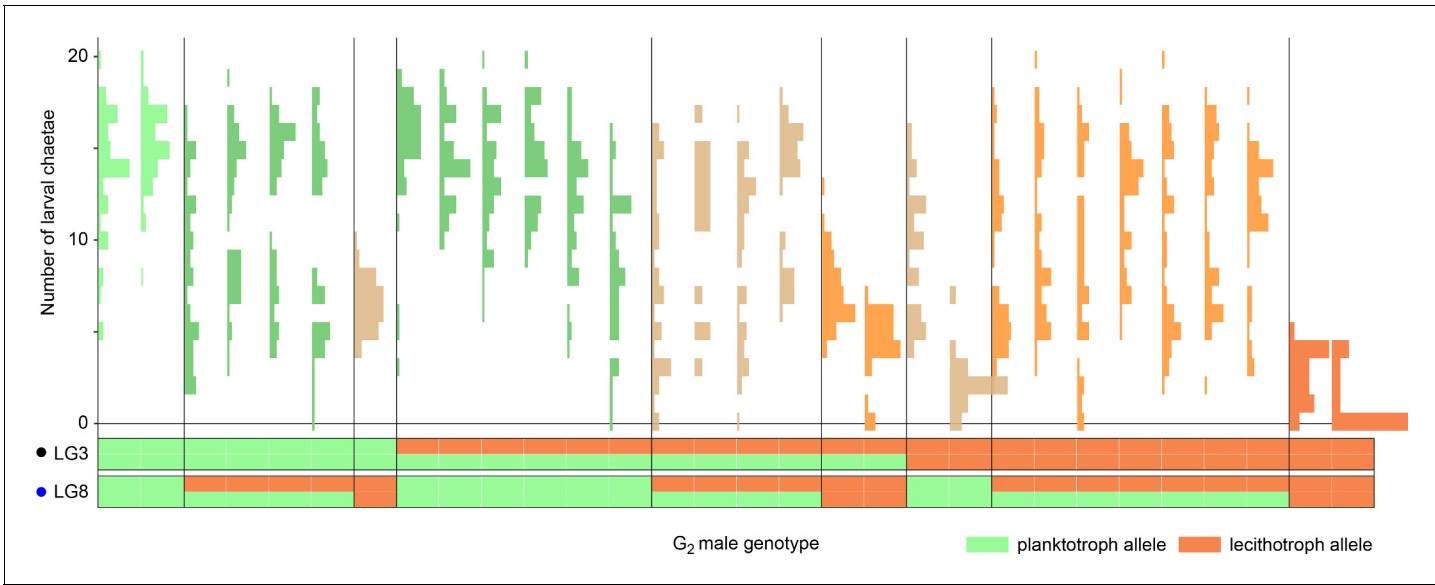

**Figure 4.** Zygotic alleles are penetrant in a planktotrophic maternal-effect background. Each histogram is the distribution of the number of chaetae in the offspring of one $G_2$ male and a planktotrophic female from the Bayonne population. Approximately 45 offspring per family were measured. Each section of the plot bounded by vertical lines represents a class of $G_2$ male genotypes at the two chaetae-number QTL. Genotypes are shown below, from doubly homozygous planktotrophic genotypes (left) to doubly homozygous lecithotrophic genotypes (right). Note that the progeny of most heterozygous $G_2$ males have multimodal distributions, demonstrating segregation of large-effect alleles. Some offspring of the males homozygous for lecithotrophic alleles completely lack chaetae, despite their planktotrophic maternal background.

DOI: https://doi.org/10.7554/eLife.37143.008

The backcross data demonstrate that zygotically-acting alleles are penetrant in planktotrophic maternal backgrounds, and thus these alleles cannot fully 'hide' in mismatched populations. However, it remains possible that variation within genotypic classes is such that an occasional larva will have a fully planktotrophic or lecithotrophic phenotype despite carrying mismatched alleles (*Zakas and Rockman, 2014*). This raises the possibility that maternal effects can allow for the accumulation of standing genetic variation in a population by masking unfit genotypic combinations from strong selection. Moreover, maternal-effect loci are sex-limited in their expression, and alleles can thus persist in populations at higher frequencies than unconditionally expressed alleles (*Wade, 1998*).

## Discussion

We find that co-adapted life-history traits are strikingly modular, where each of the phenotypes has a largely independent genetic basis with QTL occurring on different chromosomes. This suggests the lecithotrophic larval strategy evolved through the stepwise accumulation of multiple genetic changes throughout the genome. Maternally-determined larval size has an independent genetic basis from other traits that are perfectly correlated in natural populations. While the most direct evolutionary path from planktotrophy to lecithotrophy involves pleiotropic mutations that move populations diagonally across the size/shape space (*Figure 1C*), we observe that mutational effects are at right angles to one another. The lack of strong correlated effects among alleles is compatible with the evolution of lecithotrophy via a facultatively planktotrophic intermediate (*Smith et al., 2007*; *Wray, 1996*); that is, large larval size may have evolved prior to the loss of larval morphology (*Figure 1C*).

Under a broad range of assumptions, theory predicts that local adaptation in the face of ongoing gene flow should ultimately lead the causal alleles to converge at one or a few genomic loci, as opposed to an accumulation of genetic differences throughout the genome (*Kirkpatrick and Barton, 2006*; *Yeaman and Whitlock, 2011*), particularly in the context of maternal-by-zygotic epistasis (*Wade, 1998*). Our genetic results are therefore consistent with one component of the theoretical

expectations: individual traits are influenced by one or two genomic regions of large-effect, which could in turn represent clusters of underlying genes. However, the overall modularity of these phenotypes is unexpected as the loci affect discrete traits rather than the larval syndrome as a whole. Loci affecting different traits are located on different chromosomes, indicating that correlated life-history phenotypes are genetically separable. We identified six chromosomes with large effects for four traits, implying that *S. benedicti* populations can segregate $3^6$ = 729 possible genotypic combinations, most of which would produce combinations of intermediate phenotypes that are expected to be maladaptive.

Finally, we found that the evolutionary transition to lecithotrophic development requires the interaction of two genotypes, the mother and offspring. Because of this dual control of larval phenotype, the likely fitness of a zygotic allele depends on the probability it occurs in a favorable maternal genetic background. This genetic architecture provides an explanation for the phenotypic homogeneity of natural populations. The independent genetic basis of maternal and zygotic genetic effects on development shapes the accumulation of variation necessary for selection and local adaptation.

## Materials and methods

### Cross design

We crossed a single outbred Bayonne planktotrophic female to a Long Beach lecithotrophic male to generate $F_1$s. Four $F_1$ females were crossed to a single $F_1$ male to produce four half-sib $G_2$ families. We intercrossed the $G_2$ animals to generate $G_3$ families, and we crossed a subset of $G_2$ males to Bayonne females to generate backcross families.

### Phenotyping

We imaged $G_2$ individuals live so they could be reared to adulthood and crossed to produce $G_3$s. A panel of 266 $G_2$ larvae was collected within 24 hr of release from their mothers and pipetted individually onto a 0.2% agar-seawater pad on a depression slide. We imaged larvae with a Zeiss Axio Imager M2 with 20x DIC objective, and washed them into individual wells in a 24-well plate where they developed to juvenile stage, provisioned with *Dunaliella salina* algae. We measured larval size (area), chaetae number, length of the longest chaeta, and presence/absence of pronounced anal cirri from images using ImageJ (NIH).

Once $G_2$ animals reached adulthood, fed with defaunated mud, we paired them randomly for crosses. Because offspring size is proportional to egg size and has a strong additive maternal basis (*Zakas and Rockman, 2014*), we use average $G_3$ larval size as a phenotype of the $G_2$ mother. $G_3$ larvae were fixed in 4% formaldehyde overnight, transferred to PBS buffer, and mounted on slides in a long-term stable mountant. We then imaged and measured larval area as above. Fixed larvae maintained their morphology relative to live larvae when compared from the same clutch. Backcross larvae were fixed and measured in the same manner.

Raw phenotype data are included as part of an Rqtl cross object in *Supplementary file 1*, and data for backcross larvae are included in *Supplementary file 2*.

### Genotyping

For each Parent, $F_1$ and $G_2$ individual, we extracted DNA using the Qiagen DNeasy kit. Total worm DNA was sent to SNPsaurus for NextRAD genotyping-by-sequencing. NextRAD uses a primer-based approach to amplify a selection of sequences throughout the genome in order to have a reduced representation library for sequencing. PCR primers amplify genomic loci consistently between samples (*Russello et al., 2015*).

376 libraries were sequenced on Illumina HiSeq 2500 to a depth of ~20 x coverage. Reads were aligned using Samtools (*Li et al., 2009*) to a 805 MB (in fragments over 300 bp) reference genome assembly. The reference genome was constructed from Illumina Hiseq PE-100 reads of three male *S. benedicti* libraries (insert sizes of 300, 500, 800 bp) assembled with Platanus (*Kajitani et al., 2014*).

SNPsaurus aligned reads to the reference genome at 97% identity to reduce spurious alignments, and allowed only two mismatches per read to the reference. SNPs were called in vcftools (*Danecek et al., 2011*) and filtered to meet QC requirements by removing loci that had

more than two alleles or less than 10% occurrence across all samples. From more than a million scaffolds in the reference assembly, 939 contained SNPs that passed QC, resulting in 1,389 SNPs.

## Karyotyping

We imaged giemsa-stained chromosomes from cells at mitotic metaphase from laboratory-raised inbred ($F_5$-$F_8$) males and females from the Bayonne and Long Beach populations. We also imaged meiotic diakinesis in males and zygotene in females. We used a karyotyping protocol modeled closely on *Rowell et al. (2011)*. Briefly, worm tissue was incubated in 0.05% colchicine in artificial seawater for 90 min, then subjected to a hypotonic series reaching 1:4 ASW:$H_2$O over one hour. The tissue was then fixed with freshly prepared 3:1 Methanol:Acetic Acid fixative, with three fixative changes over one hour. Next, tissue was macerated on a glass slide in 60% acetic acid, using glass needles. Finally, after the slides had dried, we stained them with 5% Giemsa stain in phosphate-buffered saline. We imaged the slides on a Zeiss Axio Imager M2 with 100x objective.

Preparations from both sexes from both populations revealed a consistent karyotype of 2n = 22 and no gross differences between populations. Males are heteromorphic, carrying a distinctive Y chromosome that is substantially larger than every other chromosome. Among annelids with XY chromosomal sex determination, the Y is often larger than the X, as in species of *Hediste* (*Tosuji et al., 2004*). In *Polydora curiosa*, the only spionid previously known to have heteromorphic sex chromosomes (2n = 34 XY), the Y is larger than all other chromosomes (*Korablev et al., 1999*), as we observe in *S. benedicti*. Many other species of *Polydora* lack heteromorphic sex chromosomes (*Korablev et al., 1999*), leaving the potential homology of the large Y chromosomes in *P. curiosa* and *S. benedicti* uncertain.

Meiotic material from females of each population shows 11 bivalents at zygotene. In male meiotic material from each population, we observe 12 structures at diakinesis, including a mixture of ring and rod bivalents. We hypothesize that two of the 12 structures are X- and Y-chromosome univalents. This pattern of unpaired X and Y chromosomes at metaphase I is known among plant and insect species (*Brady and Paliulis, 2015*). *Korablev et al. (1999)* observed that the X and Y pair end-to-end at metaphase I in *Polydora curiosa*, which therefore differs from *S. benedicti*.

## Linkage map construction

Data and annotated scripts used to generate *S. benedicti* linkage maps are included in *Supplementary file 3*.

We filtered data to include only the 313 unique samples for which the mean read depth at the 1389 SNP markers was greater than 15 and for which more than 1000 SNP genotypes were called. Unfortunately, due to sample quality issues, we were unable to generate reliable genotype calls from the two $P_0$ animals or from the $F_1$ male and four $F_1$ females that parented the four $F_2$ families. We retained 47 $F_1$ and 266 $G_2$ individuals.

We tested each SNP for sex linkage by applying Fisher's exact test to genotype counts within each family and then combining the p-values using Fisher's meta-analysis. For non-sex-linked contigs that contain multiple SNPs, we manually recoded each contig, when possible, to be a single intercross marker, with inferred $P_0$ genotypes AA x BB. These inferences are straightforward given multiple SNPs per contig and our data from a panel of $F_1$s and four half-sib $G_2$ families. In some cases, individual $G_2$ families segregated haplotypes that were not all distinguishable. For example, if $P_0$ haplotypes were AB x AC, segregation in $G_2$ families descended from AC x BC $F_1$ parents is fully resolvable, but segregation from AA x BC $F_1$ parents is not, because the $P_0$ origin of A haplotypes in the $G_2$ is ambiguous. In such cases, we assigned partial genotypes in the relevant families (i.e., we treated them as dominant rather than codominant markers). However, we only recoded multi-SNP contigs for which segregation in at least one $G_2$ family was fully resolvable. Note that the assignment of genotypes to the maternal vs. paternal $P_0$ is arbitrary because we lack genotypes for those individuals. To call a marker genotype for an individual worm, we required that all SNPs in the contig marker had to have genotype calls that conform to an intact $P_0$ haplotype; this adds a layer of stringency in excluding genotypes that contain any erroneous SNP calls. Most multi-SNP contigs contain two or three SNPs, but one contained 16 SNPs spread across multiple sequencing reads. Because of missing data and genotype error, the requirement for intact $P_0$ haplotypes across 16 SNPs is too

stringent, and this contig was recoded into nine separate markers (five multi-SNP and four single SNP, based on the distribution of the SNPs across sequencing reads).

In recoding multi-SNP contigs as single markers, we identified five strains whose data showed very high rates of incompatibility with possible segregation patterns and 16 strains that showed probable pedigree errors (i.e., their genotypes were inconsistent with maternity by the $F_1$ mother shown in their records but compatible with one of the other $F_1$ mothers). These 21 strains were excluded from subsequent analysis, leaving a dataset of 292 worms distributed across five families (*Table 3*).

We inferred segregation classes for each autosomal marker and imputed genotypes for the five parents of the $G_2$ families using a composite likelihood approach. For every autosomal SNP, there are four possible $P_0$ genotype combinations (AAxBB, ABxAB, AAxAB, ABxBB), plus two for sites erroneously called as SNPs (AAxAA, BBxBB). For each SNP, we estimated likelihoods for each configuration for each family. We maximized the likelihood including a one-step error probability (i.e., AA <->AB < ->BB) allowed to range up to 10%. We next identified the segregation pattern (including the genotypes of the $F_1$ parents of the $G_2$ families) that maximizes the likelihood over the five families. This is a composite likelihood insofar as the genotype error probabilities are estimated for each family and the global pedigree likelihood derived from the appropriate combination of family likelihood estimates.

We then used four steps to build the *S. benedicti* genetic map:

1. Construction of an autosomal map from markers segregating in an intercross pattern, using Rqtl (*Broman et al., 2003*).
2. Construction of an autosomal map using the refined intercross marker dataset from step 1 and hundreds of additional non-intercross autosomal markers, using lep-MAP (*Rastas et al., 2015*; *Rastas, 2017*).
3. Addition of a sex-linked linkage-group.
4. Assignment of founder origin to each autosomal haplotype.

## Step 1: Intercross marker map

The first strategy uses data from 549 non-sex-linked SNPs that individually or in contig-groups segregate in an intercross pattern (i.e., $F_1$s are heterozygous and $G_2$s segregate 1:2:1, implying that the parental genotypes are homozygous for alternate alleles). These SNPs include 448 from multi-SNP contigs that are recoded into 193 intercross markers, plus 101 individual SNPs identified as intercross by likelihood. We excluded 17 markers that were missing data for more than 10% of individuals and three other markers that exhibited strong departures from Mendelian proportions ($p<10^{-10}$). From the resulting dataset of 274 markers and 245 $G_2$ individuals, we constructed an autosomal genetic map in Rqtl (v.1.40 – 8) following the protocol of Broman (*Broman, 2010*). 271 markers readily assembled into 10 autosomal linkage groups. We ordered the markers using 20 cycles of the *orderMarkers* function with ripple window = 7, retaining the ordering with the highest likelihood for each linkage group. Four worms exhibited a large excess of crossovers, indicative of high genotyping error rates. We removed these animals (all are Family C females), estimated the genotyping error rate as 0.013 by maximum likelihood, and reordered the markers with another 20 cycles of *orderMarkers* with error.prob = 0.013.

Some apparent crossovers are associated with single genotypes that differ from both immediate flanking markers, despite very small genetic distances. These represent probable genotyping errors.

**Table 3.** Number of individuals from each family used in mapping crosses.

| Generation | Mother | Father | Males | Females |
|---|---|---|---|---|
| $F_1$ | $P_0$ Bayonne | $P_0$ Long Beach | 24 | 23 |
| $G_2$ | $F_1$ Female A | $F_1$ Male Z | 12 | 39 |
| $G_2$ | $F_1$ Female C | $F_1$ Male Z | 15 | 63 |
| $G_2$ | $F_1$ Female F | $F_1$ Male Z | 9 | 28 |
| $G_2$ | $F_1$ Female H | $F_1$ Male Z | 22 | 57 |

DOI: https://doi.org/10.7554/eLife.37143.009

Using an error lod of 6 as a threshold for probable errors, we converted 420 (of almost 63,000) genotype calls to missing data. We then reestimated the maximum likelihood error rate as 0.004 for the remaining data and reordered the markers with 50 cycles of *orderMarkers* and error. prob = 0.004.

## Step 2. Inclusion of non-intercross markers

Returning to the original 1,389-SNP dataset, we replaced the intercross marker genotypes with those from step 1 (i.e., replacing multi-SNP contig SNPs with their inferred intercross genotypes, converting 420 genotypes to missing data, and removing the four individuals inferred to have high genotype error rates). For the remaining non-intercross SNPs, we imputed the genotypes of the five $F_1$ parents by likelihood, as described, and we used lepmap2 (*Rastas et al., 2015*) (v.0.2, downloaded March 5, 2017) to assign markers to linkage groups. The input dataset includes 288 individuals (47 $F_1$s and 241 $G_2$s in four families) and 1,111 markers. Processing steps included *ParentCallGrandparents* with XLimit = 8 and ZLimit = 8, *Filtering* with epsilon = 0.02 and dataTolerance = 0.0001, *SeparateChromosomes* with lodLimit = 15, and *JoinSingles* with lodLimit = 10 and lodDifference = 5. This analysis assigned 676 markers to 10 autosomes and 97 markers to two sex-linked chromosomes (*Figure 2*). An additional 25 markers were assigned to 10 small linkage groups (one sex-linked) with two to four markers each, and 313 markers were not assigned to linkage groups. The ten major autosomal linkage groups mapped uniquely to the ten identified using the intercross markers above at step 1.

To order the autosomal markers, we used *OrderMarkers2* in lepmap3 (*Rastas, 2017*) (v.0.1, downloaded March 5, 2017). For each linkage group, we ran 100 iterations with randomPhase = 0 and 100 iterations with randomPhase = 1, and we then repeatedly ran *OrderMarkers2* with the evaluateOrder option on the highest likelihood results for each approach to search for higher likelihood reconstructions. We then selected the result with the highest likelihood overall and retained the reconstructed genotypes (outputPhasedData = 1) for that run.

We next converted these reconstructed genotypes to conventional intercross genotypes by mapping the intercross-marker genotypes onto them and retaining the phase with the highest likelihood. We renamed the linkage groups to sort them by genetic map length and the result is Sb676. The intercross map from step 1, with linkage groups named accordingly, is Sb271.

## Step 3. Sex-linked markers

For each of the 1,389 SNPs in the original dataset, we estimated the likelihood of each possible autosomal or sex-linked segregation pattern. We identified 32 SNPs with clean X-linked segregation (AAxBY or BBxAY) and none with Z-linkage. The X-linked SNPs mapped to one major and one minor sex-linked linkage group in the lepmap analysis at step two above. The other major lepmap-defined sex-linked linkage group contained only SNPs that segregate as fixed differences between X and Y chromosomes (i.e., the maximum likelihood segregation patterns are $^XA^XA$ x $^XA^YB$ and $^XB^XB$ x $^XB^YA$). These are uninformative for the genetic map.

We next examined the multi-SNP contigs that contain any SNP whose sex-linkage p-value is less than 0.01, and where possible we recoded these SNPs as single multi-SNP markers, converting impossible genotypes to missing data, as done for autosomal markers in step 1. This resulted in a total of 26 markers (10 multi-SNP and 16 single-SNP). These markers were then ordered in Rqtl as in step 1. A group of three markers, which form a separate, minor sex-linked linkage group in our lepmap analysis at step 2, here form a distantly linked piece of the X chromosome (min.lod = 6, max. rf = 0.33). The final map, incorporating 702 markers, is Sb702.

The maps produced at each of the three steps are provided as Rqtl cross objects in *Supplementary file 1*.

## Step 4. Inference of founder haplotypes

As noted above, we were unable to recover genotypes from the founding parents of the cross or from the $F_1$ worms that contributed to the $G_2$ generation, and we therefore lack information about which autosomal alleles derive from which founder. To assign parental origins to genotypes, we tested the two possible orientations using population-genomic data. We showed previously (*Zakas and Rockman, 2015*) that slight allele frequency differentiation between Long Beach and

Bayonne populations is sufficient for population assignment. To assign the haplotypes from our cross to these populations, we generated Pool-seq libraries of each population in duplicate by pooling DNA of 25 adult females for each population, making four pool-seq libraries in total. The worms contributing to these pools were collected in 2018 from the same Long Beach and Bayonne localities as our cross founders. DNA was extracted with the Qiagen DNeasy kit and libraries were constructed with a Nextera DNA kit. These four libraries were sequenced on one lane of an Illumina NextSeq flow cell in High-Output mode. Reads were mapped to the draft reference genome, which was constructed from Bayonne individuals, using BWA and samtools (*Li and Durbin, 2009*) and variants were called using *GATK* (*McKenna et al., 2010*) with a low quality-filtering threshold of QD <1.0 and MQ <30.0.

Using this population sequence data, we calculated the likelihoods for the two possible genotype assignments for each autosome. To calculate likelihoods, we considered only SNPs that we inferred to be homozygous for alternate alleles in the founders of our cross (Step 1 above). Population allele frequencies were estimated from read counts in the pooled population sequence data. To account for the presence of zero counts of reads of some alleles in either population, which can be due to true absence or to low read coverage, we added one read to every allele count and then used these modified count proportions as population frequency estimates. In total, we used data from 110 SNPs (4 to 27 per chromosome).

For the hypothesis that founder genotype one derives from Bayonne and founder genotype two from Long Beach, we calculate the likelihood as $L(H_1)$:

P(genotype 1 | Bayonne allele frequencies) * P(genotype 2 | Long Beach allele frequencies),

assuming Hardy-Weinberg equilibrium within each population. We then calculated the likelihood of the alternative assignment ($H_2$). We report the $\log_{10}$ likelihood ratio as a statistic.

For every chromosome, one genotype assignment was favored over the alternative assignment by a factor of >300 (*Table 4*). Under the favored orientations, QTL effects are all in the expected directions (for example, large egg and short chaetae alleles derive from the lecithotrophic Long Beach founder), and these polarities are those reported in the *Supplementary file 1* genetic maps and analyzed throughout the paper.

## QTL Mapping

Annotated R scripts that reproduce all linkage mapping and phenotypic analyses in the paper are included in *Source code file 1*.

We performed interval mapping in Rqtl to identify QTL. To accommodate the family structure in the data, we performed genome scans separately in each $G_2$ family and summed the lod scores. To determine genome-wide significance thresholds, we generated a null distribution of maximum lod scores by applying the same structured genome scans to 1000 datasets in which phenotypes have been permuted among individuals within (but not among) $G_2$ families (*Churchill and Doerge, 2008*).

**Table 4.** Log10 likelihood ratio in favor of founder genotype assignments.

| Autosome | $\log_{10} (L(H_1)/L(H_2))$ |
| --- | --- |
| 1 | 9.4 |
| 2 | 23.3 |
| 3 | 2.6 |
| 4 | 13.3 |
| 5 | 9.2 |
| 6 | 22.9 |
| 7 | 11.0 |
| 8 | 10.8 |
| 9 | 26.3 |
| 10 | 4.4 |

DOI: https://doi.org/10.7554/eLife.37143.010

We used a per-trait threshold of p=0.05. All traits were mapped assuming a normal model, except for the presence or absence of anal cirri, for which we used a binary (logistic) model.

Our linkage mapping approach tests for loci that differ between the founding Long Beach and Bayonne parents of the cross, and QTL that are heterozygous in the $P_0$ founders may be overlooked. Such loci should manifest as genetic differences among the four $G_2$ families. We tested for family effects in each trait, and found no evidence for variation among the four $G_2$ families for number of chaetae (p=0.06), presence of anal cirri (p=0.70), or $G_3$ offspring size (p=0.80). We identified family effects for the length of the larval chaetae (p=0.0003) and $G_2$ size (p=$10^{-7}$).

The observed differences among $G_2$ families may be due to one or more of (1) environmental maternal effects, including familial experimental counfounders such as the exact age of full-sib larvae at the time they were measured; (2) maternal or zygotic genetic effects that segregate among the $F_2$ families due to heterozygosity in the $P_0$ founders; and (3) genetic effects that differ among $F_2$ families due to differential segregation distortion or chance allele frequency variation but not due to heterozygosity in the $P_0$s. Because $G_2$ size is a maternal-effect trait, the family effect for this phenotype can only be due to environmental maternal effects among the $F_1$ mothers or segregating maternal effect factors that were heterozygous in the $P_0$s. However, in the latter case, we expect the family effect to recur as variation among the $G_2$ families in $G_3$ offspring size. We observe no such effect, implying that the $G_2$ families do not differ in their genotype distributions at maternal-effect offspring-size loci. For larval chaetae length, we cannot exclude the possibility that some of the variation among $G_2$ families is due to QTL that were heterozygous in $P_0$s. In a model including the significant QTL (detailed below), $G_2$ family explains 5.7% of total variance in the larval chaetae length phenotype, while the two significant QTL explain 47.0%.

To model the genetics of each trait using the Rqtl *fitqtl* function, we tested for pairwise interactions between detected QTL across linkage groups and for effects of $G_2$ family. In each case we compared a model of additive and dominant within-locus effects to models that incorporated the additional terms, and we used likelihood ratio tests to determine the statistical significance of improvements in model fit. We estimate the effect size for QTL using the best-fitting model, as described in *Source code file 1*. For chaetae number and presence of anal cirri, the prefered model includes QTL effects only. For chaetae length, the model includes an effect of $G_2$ family. For $G_3$ offspring area, the model includes an interaction between two QTL. Effect estimates are reported below in *Table 4*.

We find that chaetae number and length are correlated, as in *Zakas and Rockman (2014)*, ($r^2$ = 0.22, p<0.001). The two traits share a QTL on LG3 that may act pleiotropically. However, when variation due to the LG3 QTL is removed, there is still a strong correlation between the trait residuals ($r^2$ = 0.1, p<0.001).

## Multivariate QTL scan

We analyzed the three continuous traits – mean offspring size, chaetae number, and chaetae length – in a fully multivariate QTL scan (*Knott and Haley, 2000*). The approach searches for regions of the genome with significant effects in the three-dimensional space defined by variation in these three traits. We restricted the dataset to the 149 $G_2$ females for which all three traits were scored (i.e., excluding $G_2$ worms with zero chaetae).

We fit the following model at each marker: $Y = XB + e$, where Y is the 149 × 3 matrix of phenotypes, **X** is a matrix of fixed effects, **B** is the vector of fixed-effect coefficients to be estimated, and **e** is normally-distributed residual error. The fixed effects are the intercept, $G_2$ family identity, and two vectors coding the additive and dominance variables for a single genetic marker. At each marker, we fit this model and compared it to a model with intercept and $G_2$ family as the only fixed effects, calculating a p-value from the Pillai-Bartlett statistic. We estimated thresholds for genome-wide significance by repeating the analysis on 1000 datasets in which phenotype vectors were permuted among individuals within $G_2$ families. R code that reproduces the analysis is included in *Source code file 1*.

Five QTL exceeded the genome-wide threshold for significance (p≤0.001 in each case), coinciding with the five QTL detected for these traits by univariate analysis. In a model incorporating additive and dominance effects for each QTL, five additive and two dominance effects (for the QTL on LG3 and LG6) were retained as significant (p<0.01; *Table 5*). We then estimated QTL effect sizes under the reduced model that includes only the significant effects.

**Table 5.** Estimated effects from reduced multivariate model QTL scan (in units of $G_2$ phenotypic standard deviations).

|  | Chaetae number | Chaetae length | Offspring size |
|---|---|---|---|
| Intercept | 0.18 | 0.33 | 0.13 |
| QTL3 Additive | 0.74 | 0.78 | −0.01 |
| QTL3 Dominance | −0.17 | −0.60 | −0.04 |
| QTL6 Additive | −0.01 | −0.09 | 0.55 |
| QTL6 Dominance | −0.05 | −0.19 | −0.27 |
| QTL7 Additive | 0.13 | 0.10 | 0.78 |
| QTL8 Additive | 0.57 | 0.00 | −0.13 |
| QTL9 Additive | 0.22 | 0.68 | −0.31 |
| Family C | 0.01 | −0.03 | −0.17 |
| Family F | 0.13 | −0.15 | 0.03 |
| Family H | −0.14 | −0.59 | −0.08 |

(Family A is present in the model as the reference family)

DOI: https://doi.org/10.7554/eLife.37143.011

## Analysis of backcross larvae

We tested for an effect of $G_2$ male genotype on the distribution of chaetal phenotypes in their backcross progeny. We used a linear mixed-effect model to account for the relatedness of larvae within a brood and then compared models with and without the loci implicated in each trait in the $G_2$ generation (i.e., the loci in *Table 2*). For both traits, chaetal length and number, we detected a significant effect of genotype, with effects in the expected directions. The effect size estimates indicate that the locus on LG9 does not have a significant effect on chaetal length. We note that the 30 $G_2$ males used in the backcross have unbalanced genotype representation at this locus, with 8 LL homozygotes, 22 LP heterozygotes, and 0 PP homozygotes. We are therefore poorly powered to detect an effect of this locus. R code reproducing our analyses are included in *Source code file 1*.

## Data availability

The code and experimental data generated and analyzed during this study can be found in the supplemental files.

## Acknowledgements

We thank D Tandon, I Ukegbu, R Freih, C Fayyazi, and L Jessell for laboratory assistance with maintenance and crossing of the animals, and L Noble, M Bernstein, J Yuen for helpful discussions of the manuscript. We also thank L Noble for valuable help with genome assembly and image analysis. We thank B Pernet for collecting the Long Beach worms. This research is funded by the Zegar Family Foundation, NSF grant IOS-1350926 to MVR, NIH grant GM108396-02 to CZ, and an NYU Biology Master's Research Grant to ADK.

## Additional information

### Funding

| Funder | Grant reference number | Author |
|---|---|---|
| National Science Foundation | IOS-1350926 | Matthew V Rockman |
| National Institutes of Health | GM108396 | Christina Zakas |
| Zegar Family Foundation |  | Matthew V Rockman |
| New York University | Biology Master's Research Grant | Alex D Kay |

The funders had no role in study design, data collection and interpretation, or the decision to submit the work for publication.

## Author contributions

Christina Zakas, Conceptualization, Resources, Data curation, Software, Formal analysis, Funding acquisition, Investigation, Visualization, Methodology, Writing—original draft, Writing—review and editing; Jennifer M Deutscher, Investigation, Visualization; Alex D Kay, Funding acquisition, Investigation; Matthew V Rockman, Conceptualization, Resources, Data curation, Software, Formal analysis, Supervision, Funding acquisition, Investigation, Visualization, Methodology, Writing—review and editing

## Author ORCIDs

Matthew V Rockman (iD) http://orcid.org/0000-0001-6492-8906

## Decision letter and Author response

Decision letter https://doi.org/10.7554/eLife.37143.018
Author response https://doi.org/10.7554/eLife.37143.019

## Additional files

### Supplementary files

• Supplementary file 1. R workspace file containing *S. benedicti* genetic maps and $G_2$ phenotype data. The data are stored as rqtl (*Broman et al., 2003*) cross objects.
DOI: https://doi.org/10.7554/eLife.37143.012

• Supplementary file 2. csv file containing phenotype and genotype data for Bayonne backcross larvae.
DOI: https://doi.org/10.7554/eLife.37143.013

• Supplementary file 3. Compressed directory containing data and scripts used to generate the *S. benedicti* genetic maps presented in *Supplementary file 1*. The directory includes R scripts for each of the four steps of map construction detailed in the Materials and Methods, and a perl script required for step 2. The directory also includes data files called by the R scripts.
DOI: https://doi.org/10.7554/eLife.37143.014

• Source code 1. R script file containing the annotated workflow underlying all genetic mapping and phenotypic analyses presented in the manuscript. This script makes use of the data in *Supplementary file 1* and *Supplementary file 2*.
DOI: https://doi.org/10.7554/eLife.37143.015

• Transparent reporting form
DOI: https://doi.org/10.7554/eLife.37143.016

### Data availability

All code and data generated and analyzed during this study can be found in the supplemental files.

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
