## [Decision Letter]

Thank you for submitting your article "Decoupled maternal and zygotic genetic effects shape the evolution of development" for consideration by *eLife*. Your article has been reviewed by three peer reviewers, including Magnus Nordborg as the Reviewing Editor and Reviewer #1, and the evaluation has been overseen by Diethard Tautz as the Senior Editor. The following individual involved in review of your submission has agreed to reveal their identity: Kristin Tessmar (Reviewer #2).

The reviewers have discussed the reviews with one another and the Reviewing Editor has drafted this decision to help you prepare a revised submission.

Summary:

This paper investigates the genetics of a major life-history dimorphism – development via a morphologically distinct larval form vs. direct development – in a marine worm. Rather than "super-genes" determining life-history strategy, separate genetic loci with different modes of inheritance were found to influence egg size and larval morphology.

This is a substantial and well-written contribution addressing a biological question of broad interest.

Essential revisions:

There were two major issues that could lead to over-interpretation of the results. First, fitness effects are implied but not measured, and this is not always clear from the writing. A hypothetical fitness landscape is shown in Figure 1B, and it is clearly stated in the figure legend that this landscape is hypothetical. However, at several other points in the text, sentences are worded in a way that implies that fitness may have actually been measured, rather than inferred based on direction of phenotypic effects. This could be fixed by more cautious phrasing. For example:

In Abstract, original text: "The fitness of zygotic alleles depends upon their maternal background, creating a positive frequency-dependence that may homogenize local populations."

More conservative wording: "The likely fitness effects of zygotic alleles depends upon their maternal background, creating a positive frequency-dependence that may homogenize local populations."

Even more conservative wording: "The phenotypic effects of zygotic alleles on a hypothetical fitness landscape depends upon their maternal background, creating a positive frequency-dependence that may homogenize local populations."

In the Discussion, original text: "We identified six chromosomes with large effects for four traits, implying that *S. benedicti* populations can segregate 729 possible genotypic combinations, most of which are maladaptive."

More conservative wording: "We identified six chromosomes with large effects for four traits, implying that *S. benedicti* populations can segregate 729 possible genotypic combinations, most of which would produce combinations of intermediate phenotypes that are likely to be maladaptive."

In the Discussion, original text: "Finally, we found that the evolutionary transition to lecithotropic development requires the interaction of two genotypes, the mother and offspring. The fitness of a zygotic allele depends on the probability it occurs in a favorable maternal genetic background"

More conservative: "Finally, we found that the evolutionary transition to lecithotropic development requires the interaction of two genotypes, the mother and offspring. Because of this dual control of larval phenotype, the likely fitness effects of a zygotic allele depends on whether it occurs in a favorable maternal genetic background." etc.

The second point is more difficult to address. Due to failed genotyping, the genotypes of the P0 grandparents of the existing cross, or the F1 parents used to make F2s, are not known. Thus the direction of the allelic effects in the cross is actually not known, although the paper is generally written as if they went in the expected direction (again, the information is in the paper, but it is very easy to miss on a first reading). This is not a trivial point, and it affects the conclusions of the paper. For example, the difficulty contrasting alleles would have invading different genetic backgrounds is less of an issue if existing planktotrophic and lecithotropic populations already harbor lots of alleles with non-concordant phenotypic effects.

This issue could be addressed by carrying out additional genotyping to try to confirm whether the planktotrophic and lecithotropic alleles segregating in the cross do in fact come from either the planktotropic and lecithotropic grand parental populations. Note that the additional genotype calls would only be in the regions harboring key QTLs, rather than on a whole genome basis. With a focused genotyping effort, it may be possible to determine these key genotypes on limited or poor quality material still available from the original grandparents. Alternatively, it may be possible to infer the likely grandparental genotypes in the key QTL regions, based on the differential frequency of linked markers found in additional individuals sampled from the starting populations.

If you cannot produce these data, a thorough rewrite will be required to make sure all claims are supported, and the manuscript will be reviewed again.

Less substantially, please clarify in what sense Figure 4 shows maternal "masking", and in general improve description, i.e. clearly label what exactly is the offspring of one cross, how many individuals were scored for each chaetae number.…

---

## [Author Response]

Essential revisions:There were two major issues that could lead to over-interpretation of the results. First, fitness effects are implied but not measured, and this is not always clear from the writing. A hypothetical fitness landscape is shown in Figure 1B, and it is clearly stated in the figure legend that this landscape is hypothetical. However, at several other points in the text, sentences are worded in a way that implies that fitness may have actually been measured, rather than inferred based on direction of phenotypic effects. This could be fixed by more cautious phrasing. For example:In Abstract, original text: "The fitness of zygotic alleles depends upon their maternal background, creating a positive frequency-dependence that may homogenize local populations."More conservative wording: "The likely fitness effects of zygotic alleles depends upon their maternal background, creating a positive frequency-dependence that may homogenize local populations."Even more conservative wording: "The phenotypic effects of zygotic alleles on a hypothetical fitness landscape depends upon their maternal background, creating a positive frequency-dependence that may homogenize local populations."In the Discussion, original text: "We identified six chromosomes with large effects for four traits, implying that S. benedicti populations can segregate 729 possible genotypic combinations, most of which are maladaptive."More conservative wording: "We identified six chromosomes with large effects for four traits, implying that S. benedicti populations can segregate 729 possible genotypic combinations, most of which would produce combinations of intermediate phenotypes that are likely to be maladaptive."In the Discussion, original text: "Finally, we found that the evolutionary transition to lecithotropic development requires the interaction of two genotypes, the mother and offspring. The fitness of a zygotic allele depends on the probability it occurs in a favorable maternal genetic background"More conservative: "Finally, we found that the evolutionary transition to lecithotropic development requires the interaction of two genotypes, the mother and offspring. Because of this dual control of larval phenotype, the likely fitness effects of a zygotic allele depends on whether it occurs in a favorable maternal genetic background." etc.

We have made all of the wording changes that the reviewers suggest, and we have modified other references to fitness to make clear that we expect intermediate phenotypes to be suboptimal based on theory but that we have not measured fitness.

The second point is more difficult to address. Due to failed genotyping, the genotypes of the P0 grandparents of the existing cross, or the F1 parents used to make F2s, are not known. Thus the direction of the allelic effects in the cross is actually not known, although the paper is generally written as if they went in the expected direction (again, the information is in the paper, but it is very easy to miss on a first reading). This is not a trivial point, and it affects the conclusions of the paper. For example, the difficulty contrasting alleles would have invading different genetic backgrounds is less of an issue if existing planktotrophic and lecithotropic populations already harbor lots of alleles with non-concordant phenotypic effects.This issue could be addressed by carrying out additional genotyping to try to confirm whether the planktotrophic and lecithotropic alleles segregating in the cross do in fact come from either the planktotropic and lecithotropic grand parental populations. Note that the additional genotype calls would only be in the regions harboring key QTLs, rather than on a whole genome basis. With a focused genotyping effort, it may be possible to determine these key genotypes on limited or poor quality material still available from the original grandparents. Alternatively, it may be possible to infer the likely grandparental genotypes in the key QTL regions, based on the differential frequency of linked markers found in additional individuals sampled from the starting populations.If you cannot produce these data, a thorough rewrite will be required to make sure all claims are supported, and the manuscript will be reviewed again.

The reviewers make an important point about the direction of the QTL effects, and we have generated new data to address it. Unfortunately, we were not able to add genotypes from the original parents due to the limited, degraded DNA. We instead used a population genetic approach to infer the source population for the alternative haplotypes in our genetic map. We collected 25 individuals from each population and used Pool-seq methods to estimate allele frequencies in each. We then calculated the likelihood of alternative population assignments for each founder chromosome. We find strong support from the allele frequencies for a specific population assignment for each founder haplotype, and the inferred assignments are consistent with the directional effects described in our manuscript. This has important implications for interpretation because it demonstrated that the QTL effects are acting in the direction we expected; they are complementary not contrasting effects. We have added the details of our analysis to the mapping part of the Materials and methods section and we provide the underlying data and analysis scripts as part of Supplementary file 3.

Less substantially, please clarify in what sense Figure 4 shows maternal "masking", and in general improve description, i.e. clearly label what exactly is the offspring of one cross, how many individuals were scored for each chaetae number.

We changed the figure legend to clarify the cross distributions. The offspring in these 30 crosses have a different maternal background from the rest of our dataset (which have F_1_ mothers), but we see the same pattern where the number of chaetae depends on the zygotic genotype at the two QTL. At the same time, we would predict that because of the maternal planktotrophic contribution that all of these offspring would develop some chaetae, but some that have lecithotrophic paternal alleles do not, demonstrating the incomplete zygotic penetrance of this trait.